# Biological Significance and Therapeutic Promise of Programmed Ribosomal Frameshifting

**DOI:** 10.3390/ijms26031294

**Published:** 2025-02-03

**Authors:** Miora Bruna Marielle Ramamonjiharisoa, Sen Liu

**Affiliations:** 1Cooperative Innovation Center of Industrial Fermentation (Ministry of Education & Hubei Province), Key Laboratory of Fermentation Engineering (Ministry of Education), Wuhan 430068, China; miorabruna@gmail.com; 2Hubei Key Laboratory of Industrial Microbiology, National “111” Center for Cellular Regulation and Molecular Pharmaceutics, School of Life and Health Sciences, Hubei University of Technology, Wuhan 430068, China

**Keywords:** programmed ribosomal frameshifting, translational regulation, polyamine, eIF5A

## Abstract

Programmed Ribosomal Frameshifting (PRF) is a mechanism that alters the mRNA reading frame during translation, resulting in the production of out-of-frame proteins. PRF plays crucial roles in maintaining cellular homeostasis and contributes significantly to disease pathogenesis, particularly in viral infections. Notably, PRF can induce immune responses in the SARS-CoV-2 mRNA vaccine, further extending its biological significance. These multiple aspects of PRF highlight its potential as a therapeutic target. Since PRF efficiency can be modulated by cellular factors, its expression or silencing is context-dependent. Therefore, a deeper understanding of PRF is essential for harnessing its therapeutic potential. This review explores PRF biological significance in disease and homeostasis. Such knowledge would serve as a foundation to advance therapeutic strategies targeting PRF modulation, especially in viral infections and vaccine development.

## 1. Introduction

Gene translation is a fundamental process where genetic information is converted from mRNA to amino acids through ribosomal decoding. The decoding process begins at the open reading frame (ORF) upon the recruitment of the initiator tRNA carrying methionine (Met-tRNAi) to the start codon (AUG) on the mRNA. Trans-acting factors ensure the correct positioning of the initiator codon on the ribosome [1]. During translation elongation, the ribosome moves along the mRNA by three nucleotides (a codon) at each step, appending the corresponding amino acids to the extending polypeptide chain. Elongation factors facilitate the binding of aminoacyl-tRNAs to the ribosome, ensuring the correct codon–anticodon pairing and maintaining the correct reading frame. These factors use guanosine triphosphates as an energy source to drive the elongation process [2]. Upon reaching a termination codon, release factors recognize the codon and promote the release of the newly synthesized polypeptide. Subsequently, the ribosome dissociates into its small subunit and large subunit, preparing for another round of translation [1]. This precise process ensures the correct in-frame translation of an mRNA sequence into the corresponding polypeptide. 

Programmed Ribosomal Frameshifting (PRF) is a regulated event that disrupts the normal translation process. It causes the ribosome shifting to an alternative reading frame at a specific codon in the mRNA sequence. This shift results in the production of an out-of-frame protein [3]. PRF plays a critical role in gene expression, protein synthesis, and cellular homeostasis. PRF is also implicated in disease pathogenesis, particularly in viral infections [4]. The efficiency of PRF can be modulated by cellular factors, such as polyamine and translational factors [5,6,7,8,9,10]. Additionally, frameshift proteins resulted from PRF can elicit immune responses, as observed in the SARS-CoV-2 IVT mRNA vaccine [11].These properties make PRF a promising target for therapeutic intervention, especially in vaccine development and antiviral strategies. This review focuses on the molecular mechanisms of PRF, its dual roles in maintaining cellular homeostasis, and contributing to pathogenesis. We discuss the regulation of PRF by polyamines and the eukaryotic translation initiation factor 5A (eIF5A). We also explore PRF’s therapeutic potential by reviewing advancements targeting this mechanism. Altogether, this review highlights the relevance of PRF in RNA-based treatments, particularly in antiviral strategies and vaccine development.

## 2. Types and Mechanisms of Programmed Ribosomal Frameshifting

PRF is activated by specific cis-acting signals within mRNA [3], leading to nucleotide skipping in either upstream or downstream of the in-frame codon (Figure 1). PRF typically happens when a codon is slow to be decoded (also called a hungry codon) due to the low abundance of its corresponding tRNAs [12]. PRF classification is based on the direction and number of nucleotides shifted, with −1 and +1 frameshifting being the most common, while −2 PRF is less frequent [13,14]. Importantly, there is also a possible conversion between these different classes of PRF. This dynamic interconversion underscores the malleability and complexity of PRF.

The PRF event typically involves a heptameric slippery sequence that disrupts the standard translation process, causing the ribosome to pause and facilitating tRNA slippage. Frameshift stimulatory elements (FSEs) form secondary structures like pseudoknots and modulate the efficiency of frameshifting by creating energetic barriers for the ribosome. These elements constitute the cis-acting complex that strictly regulates frameshifting efficiency. Minor changes within this complex can significantly affect the frameshifting outcome.

In +1 PRF, the elongating ribosome is directed to a kinetic slippage of the Peptidyl-site (P-site) tRNA in the 3′ direction. As exemplified by the human ornithine decarboxylase antizyme 1 (OAZ1) mRNA, the ribosome shifts forward, bypassing a stop codon to initiate an alternative ORF2 that encodes the full-length OAZ1 [15]. This process is regulated by cellular polyamines, which bind to the ribosome and influence frameshifting efficiency [5]. The conserved sequence elements in OAZ1 mRNA include the slippery sequence UCC UGA U, a 5′ FSE consisting of eleven nucleotides from the last four codons of ORF1 (UGG-UGC-UCC-UGA) [16], the nucleotide after the stop codon with typically C or U in a conserved pyrimidine-rich sequence (UCCCU) [16,17], and a 3′ pseudoknot formed by nucleotides 193 to 269 [17,18]. These elements influence mRNA-tRNA interactions in the P-site, leading to a quadruplet translocation at UCCU [19].

−1 PRF is common in viral genomes. This PRF allows for the expansion of genetic information by shifting the reading frame backward. The cis-acting signal is a slippery sequence formed by a heptameric motif X XXY YYZ. XXX represents three identical nucleotides, YYY are mostly AAA or UUU, and Z can vary. The frameshifting is induced by the YYZ “hungry” codon in the Aminoacyl-site (A-site), then the tRNA reading frame is shifted upward to YYY in the 5′ direction [20]. Another possibility is that the ribosome may pause at the slippery sequence, causing the A- and P-site tRNAs to slip back by one base together, resulting in a −1 frameshift [21]. The conserved 3′ FSE is either a stem loop, a hairpin, or a pseudoknot. It consists of a sequence of 1 to 12 nucleotides following the slippery sequence [20,21].

−2 PRF is less common compared to −1 and +1 PRFs [22]. One example is in the translation of the non-structural protein 2 (NSP2) of arteriviruses [13]. The most common slippery sequence of this PRF is RG GUU UUU, where R can be G or A. This sequence can trigger both −1 and −2 PRF, producing two variants of NSP2 [22]. The −2 PRF signal induction arises from the 0 frame UUU or GUU. It allows the total repairment of the A-site or a partial repairment of the P-site, respectively. The shift occurs by a tandem slippage of the ribosome-bound tRNAs [23]. The frameshifting efficiency is regulated by upstream and downstream FSEs. The upstream FSE involves a complex formed by a host poly-C-binding protein and the viral NSP1β protein. This complex binds to the mRNA near the slippery sequence and likely creates a roadblock that influences ribosome movement, enhancing PRF efficiency. The 3′ FSE contains a C-rich motif (CCCANCUCC) located 10 nucleotides downstream of the slippery site, further modulating the frameshifting process [14,22].

Conversion between different classes of PRF can occur depending on changes of the cis-acting signals [15,24]. In arteriviruses, as mentioned above, the same slippery sequence can develop both −1 and −2 PRF from the same frame 0. This frame 0 is either GUU or UUU in the P-site or A-site, respectively. A study showed that the spacer between the slippery sequence and the 3′ FSE (specifically, an RNA secondary structure) determines the occurrence of both PRFs. The levels of the −1 and the −2 PRFs change when the spacer is shortened [25].

In OAZ1, the +1 PRF can be converted to a −2 PRF too. The change depends on host context. The +1 PRF in the expression of OAZ1 is conserved from yeast and mammals [19]. However, when the mammalian OAZ1 was introduced in *Saccharomyces cerevisiae*, the same reading frame underwent −2 PRF by reading CC in the slippery sequence UCC UGA U twice [15]. This conversion is likely due to the slippage of mRNA and re-pairing with the tRNA in the P-site. The downstream pseudoknot stimulates frameshifting differently in yeast and mammalian systems. For instance, PRF efficiency increases 30-fold in yeast compared to a 2.5-fold increase in reticulocyte lysates. The spacer length between the shift site (from UCC) to the pseudoknot is also a key determinant of the conversion. When the spacer length is extended by three nucleotides, the efficiencies of +1 and −2 frameshifting become equal [15]. 

Conversion of −1PRF to −2 or +1 is also possible. For example, in coronavirus infectious bronchitis virus, the −1 PRF is converted into −2 or +1 depending on the slippery sequence and pseudoknot [24]. The monotonous UUU in the slippery sequence UUUAAAC is a key element favorizing −2/+1 conversion. The efficiency of −2/+1 frameshifting increases (up to 21%) when the slippery sequence is extended to eight consecutive U bases. The pseudoknot stability also can influence the frameshift direction [24]. 

## 3. Diverse Biological Functions of PRF

PRF plays a critical role in the regulation of gene expression. Studies showed that PRF exhibits dual facets of regulation on biological systems, either supporting physiological cellular functions [26] or contributing to disease progression [27] (Figure 2).

### 3.1. PRF for Maintaining Homeostasis

PRF plays pivotal roles in maintaining cellular homeostasis, such as polyamine homeostasis [5,28]. Polyamines are crucial for cell viability and growth. The cellular polyamine level is regulated by OAZ1, an inhibitor of ornithine decarboxylase (ODC), the rate-limiting enzyme in polyamine biosynthesis. The +1 PRF of OAZ1 mRNA is stimulated by polyamines, serving as a critical feedback mechanism in polyamine synthesis regulation. When polyamine levels are elevated, the +1 PRF results in the expression of full-length OAZ1, which inhibits ODC and extracellular polyamine uptake [29]. Conversely, low polyamine levels lead to the expression of a truncated, non-functional OAZ1, allowing for increased polyamine synthesis and uptake [30]. This PRF efficiency is approximately 26% in mammals [5]. It is governed by a slippery sequence and FSEs with specific secondary structures. The slippery sequence for OAZ1 +1 PRF varies across species. In yeast and fungi, it is UUU UGA C, and in humans and most metazoans, it is UCC UGA U [31]. The downstream FSE typically includes two stem loops about 16 nucleotides starting with a conserved tetranucleotide [32]. The upstream FSE involves a module A of 6 to 7 nucleotides directly upstream the slippery site, followed by a sequence corresponding to a nascent peptide of 11 amino acids [5,33].

### 3.2. PRF for Ensuring Protein Stoichiometry

PRF plays a critical role in maintaining the stoichiometric balance of viral proteins, which is essential for viral replication and pathogenicity [4]. In retroviruses like the human immunodeficiency virus (HIV), −1 PRF is crucial for the translation of the Gag-Pol polyprotein, because it ensures the proper ratio of structural and non-structural proteins. The slippery sequence is a U UUU UUA stretch situated within the gag/pol overlap ORFs. The gag ORF encodes viral structural proteins, whereas the pol ORF encodes enzymes such as reverse transcriptases, integrases, and proteases [34]. The frameshift is initiated by a tandem slippage in the 5′ direction of both peptidyl-tRNA-Phe and aminoacyl-tRNA-Leu. It can also occur by a single slippage of peptidyl-tRNA-Phe [35]. The frameshifting efficiency is approximately 5–10%, and variations in this efficiency impact viral replication [36]. A 3′ downstream stem loop formed by a highly conserved GGG sequence in the ribosomal A-site stimulates PRF efficiency. This GGG sequence behaves as a “hungry” codon by slowing translation kinetics and significantly favorizing frameshifting [37]. Alterations to this codon decrease PRF efficiency [35].

In West Nile virus (WNV), −1 PRF affects the balance of non-structural and structural proteins, thereby affecting viral replication and assembly. This frameshift occurs in the slippery sequence CCU UUU CAG during the decoding of NSP1 when tRNA-Gln slips in the U-rich region [38]. The frameshift happens in about 50% of translation events [39]. It results in an elongated NSP1 protein [40], halting the synthesis of subsequent NSPs such as NSP2 [41]. A pseudoknot structure within the NSP2 mRNA serves as a key stimulatory element for this PRF event [41]. The role of this PRF in viral replication is unclear [41,42]. When the elongated NSP1 was inhibited, the Japanese Encephalitis Virus strain SA14-14-2, an attenuated vaccine by the World Health Organization, showed reduced neurovirulence and neuroinvasiveness in mice [39]. Other studies suggested that this PRF may facilitate virus replication in birds and mosquitoes [42]. However, in vitro cell lines study found no impacts of PRF and the elongated NSP1 on viral replication [43].

Porcine Reproductive and Respiratory Syndrome Virus (PRRSV) causes respiratory issues in pigs and reproductive failures in sows [44]. A −1 PRF event occurs at the overlap of ORF1a and ORF1b. It is initiated at the slippery sequence U UUA AAC and directs the translation of ORF1b encoding essential enzymatic proteins [45]. This PRF has an efficiency of 15–20% [46]. The FSE includes a downstream pseudoknot located 5 nucleotides from the slippery sequence [45]. Additionally, PRF efficiency increases with the progression of infection [45].

### 3.3. PRF for Producing Pathogenic Proteins

PRF is essential for generating viral proteins that contribute to pathogenesis. In PRRSV, −2 and −1 PRF during NSP2 translation yields an elongated version NSP2TF and a truncated version NSP2N, respectively. Two thirds of the N-terminal sequence are same in these two variants. Both variants possess protease activities including deubiquitination and deISGylation, contributing to immune antagonism [45]. The frameshifting product NSP2TF exhibits stronger inhibition of interferon β and α signaling and reduces NK cell cytotoxicity, highlighting its role in immune evasion [14,47]. NSP2TF also anchors to perinuclear membranes and targets the exocytic pathway by stabilizing structural proteins GP5 and M [48]. This regulatory mechanism is also observed in other viruses like the encephalomyocarditis virus [49]. It suggests an adaptive strategy to optimize protein production at various infection stages to effectively suppress host responses. The efficiencies of −2 and −1 PRF in this case are 20% and 7%, respectively [45]. 

In SARS-CoV-2, −1 PRF is essential for translating the ORF1b polyprotein. ORF1b encompasses the viral RdRp and NSP12, two proteins essential for viral replication and later stages of infection. Instead of stopping at an in-frame stop codon at the end of ORF1a, −1 PRF occurs over the slippery sequence U UUA AAC. This PRF efficiency is influenced by upstream and downstream RNA structures. The upstream structure contains a 5′ attenuator loop [50] that inhibits PRF by reducing ribosomal availability [21]. The downstream structure is located five codons beyond the frameshift site. It is a pseudoknot that features a complex three-dimensional fold with two stems separated by a bulge of 8-nucleotides [50]. Such structure induces ribosomal pausing, facilitating tRNA realignment for the backward shift in the reading frame. This PRF’s efficiency ranges from 25–75% [51]. The pathogenicity of the products from this PRF is debated. Some studies suggested that they act as interferon antagonists, as evidenced by low IFN levels in the serum of COVID-19 patients [52,53,54], while others proposed that they modulate interferon production [55,56].

Frameshift proteins can elicit immune responses, as demonstrated in the example of SARS-CoV-2 in vitro transcribed (IVT) mRNA vaccines. In IVT mRNA vaccines, the use of modified ribonucleosides, such as N1-methylpseudouridine is a common strategy to counteract mRNA instability and innate immunogenicity [57]. An in vitro study revealed that this modification in a specific type of SARS-CoV-2 vaccine unexpectedly induces a +1 PRF. This frameshifting event leads to the production of frameshift proteins, which could trigger immune responses and alter the anticipated immune profile [11]. 

## 4. Molecular Regulation of PRF

The regulation of PRF extends beyond simple cis-acting elements to more complicated mechanisms (Figure 3). These mechanisms involve dynamic interactions between various macro-molecules and/or small metabolites [4].

### 4.1. PRF Regulation by Polyamines

Polyamines, including putrescine (PUT), spermidine (SPD), and spermine (SPM), are small polycationic molecules that play a significant role in regulating PRF across various organisms [5]. Their positive charges are believed to neutralize the negative charge repulsion between mRNA and rRNA [58]. 

The polyamine-dependent PRF mechanism is conserved across yeast to mammals, particularly in the decoding of OAZ1 mRNA [5,59,60]. High polyamine levels stimulate this +1 PRF to ensure the translation of full-length OAZ1 [5]. Polyamines bind to the nascent OAZ1 polypeptide within the upstream FSE in the mRNA sequence [5]. This binding neutralizes rRNA repulsion. Thus, when the ribosome pauses at the 0-frame UGA stop codon, tRNA can slip in the P-site more easily [3]. Consequently, this mechanism prevents ribosome stalling on OAZI mRNA. 

The retrotransposon Ty1 PRF in *Saccharomyces cerevisiae* is also regulated by polyamines. This +1 PRF occurs during the translation of the pol protein to ensure the retransposition [6]. When a ribosome counteracts the rare AGG-Arg codon in the slippery sequence CUU AGG C, a leucyl-tRNA slips from CUU to UUA. The frameshifting efficiency in this case is approximately 7% [7]. Deficiency in SPD combined with an increase in PUT stimulates this PRF, inhibiting the retrotransposition [8] and vice-versa [9]. This indicates that a specific PUT/SPD ratio is crucial for regulating Ty1 retrotransposon PRF [6].

### 4.2. PRF Regulation by Hypusinated eIF5A

The elongation factor eIF5A plays essential role in eukaryotic translation initiation, elongation, and termination. Hypusination is critical for its function and is polyamine-dependent [61,62,63]. This relationship with polyamines confers to eIF5A its regulatory function toward PRF.

In OAZ1 PRF, hypusinated eIF5A regulates PRF through its role in the synthesis of antizyme inhibitor 1 (AZIN1) [64]. AZIN1 mRNA has an upstream conserved coding region with a non-AUG start codon. Translation of this region is favorized by elongating ribosomes pausing in a conserved Pro-Pro-Trp motif [65]. This pausing impairs the induction of translation in the downstream AUG start codon and represses AZIN1 synthesis [65]. eIF5A regulates this ribosome pausing depending on polyamines availability for its hypusination [65]. Hypusinated eIF5A promotes efficient translation of AZIN1 mRNA by preventing ribosomal pausing, while reduced hypusination leads to increased pausing [65]. Since AZIN1 outcompetes ODC binding to OAZ1, its eIF5A-dependant synthesis regulates polyamine synthesis and OAZ1 PRF consequently [62]. 

A similar regulation occurs in the polyamine transporter HOL1 of *Saccharomyces cerevisiae*. eIF5A ensures the translation termination at a Pro-Ser-stop motif in an upstream ORF on the HOL1 transporter mRNA under low polyamine conditions [6,65]. Therefore, eIF5A function contributes to the increase in cellular polyamine uptake and further regulates OAZ1 PRF. 

eIF5A also regulates the Ty1 retrotransposon PRF with similar mechanism as described above. Hypusinated eIF5A increases the PUT/SPD ratio, thus promoting the +1 PRF of Ty1. The proximity of the in-frame stop codon to the slippery sequence is crucial for the eIF5A regulation of PRF in yeast [6].

Hypusinated eIF5A also regulates −1 PRF efficiency in SARS-CoV-2 [6]. Depletion of eIF5A in human cells impairs this PRF. Similar to the case of Ty1 PRF above, this dependency is likely due to the close proximity of the in-frame stop codon to the slippery sequence. The distance between them is less than one ribosomal footprint upstream (about 30 nucleotides) [6]. When this distance is larger as in the example of other betacoronavirus, eIF5A regulation is impaired [21]. This configuration is essential for preventing trailing ribosomes from encountering the slippery sequence while it is engaged by a leading ribosome [6,10].

## 5. Therapeutic Potential of PRF

PRF plays a dual role in maintaining cellular homeostasis and in viral infections, presenting a promising target for therapeutic interventions. Targeting PRF regulation can combat diseases and infections through modulation of frameshifting efficiency. Upstream FSEs typically stimulates frameshifting efficiency while downstream FSEs inhibits frameshifting [66,67,68]. Frameshift proteins could also stimulate immune responses, underscoring PRF’s role in vaccine development (Figure 4).

### 5.1. Potential of PRF Inhibition

PRF is crucial for viral replication, making it a viable target for antiviral therapies. Targeting the 3′FSEs to inhibit PRF efficiency is one way to perturb viral replication. Disrupting proteins equilibrium is another approach. These strategies hold a promise to advance antiviral therapy. 

The use of antibiotics demonstrates some successes in targeting the three-stems pseudoknot structure [50]. This structure is commonly found in bacterial riboswitches. That might explain the sensitivity of these structures to antibiotics [69]. Various antibiotics have underscored antiviral potential in this regard. 

Geneticin, for instance, is an RNA-binding antibiotic belonging to the aminoglycoside class. It can inhibit −1 PRF efficiency in multiple variants of SARS-CoV-2 by binding to the 3′ pseudoknot. This binding leads to the inhibition of viral replication and protein expression. In silico modeling of the pseudoknot cryo-EM structure suggests that the binding pocket is likely situated between stem 1 and stem 2 (Site 1), in the junction site between stem 2 and stem 3 (Site 2), or at the beginning of stem 2 (Site 3) [70,71]. Among them, Site 1 appears to have the highest potential for binding. The reason relies in the presence of three key PRF-inducing nucleotides (U45, A74, and U75) in this site. Geneticin exhibits sustained antiviral activity without inducing significant resistance. However, its use in therapy against SARS-CoV-2 is risky because it requires high concentrations (in the micromolar range) to be active. It also contains an N-nitroso group, raising concerns about potential carcinogenicity [71]. 

Guanidinoneomycin B is another aminoglycoside derivative of neomycin B that can bind to the HIV-1 downstream stem loop. The binding significantly increases the stability of the −1 PRF by increasing the melting temperature of the frameshift site. This results in a stoichiometry imbalance between gag and pol proteins, decreasing the rate of viral replication. NMR spectroscopy revealed that the binding between the antibiotic and the stem loop form a 1:1 complex within a highly electronegative pocket. This interface is formed by seven G-C pairing, alongside a structured ACAA tetraloop, where guanidinoneomycin B fits. However, this compound lacks specificity. It can bind to other RNA motifs, whether or not they are engaged with such stem loop [72]. 

Unlike these two antibiotics mentioned above, merafloxacin belongs to the fluoroquinolone antibacterial class. It can inhibit −1 PRF in SARS-CoV-2 by disrupting the 3′ pseudoknot structure. This PRF inhibition impedes viral replication in Vero E6 cell lines [73]. Moreover, this inhibition is resistant to sequence mutations within the pseudoknot and is reproducible in other beta coronaviruses [73]. Merafloxacin likely binds to the same binding pocket as Geneticin and shares the same mechanism of action [71].

Peptidyl transferase inhibitors target PRFs differently by interfering with the ribosome pausing in the peptidyl transfer center. For example, anisomycin alters −1 PRF in L-A virus. It precisely inhibits the binding of aminoacyl-tRNA to the A-site. Such inhibition is only possible when the A and P-site are occupied by cognate tRNAs. Thus, in +1 PRF where the A-site is vacant, peptidyl transferase inhibition is impaired. The decrease in −1 PRF results in a viral loss through imbalance between Gag/Pol proteins [74].

RNA-binding proteins such as Annexin A2 and RG501 can also disrupt viral PRFs by their binding in the 3′ pseudoknot. Annexin A2 is a natural mRNA binding protein. It binds to the 3′ pseudoknot in infectious bronchitis virus and inhibits virus replication. Any mutations can impair the complex. The −1-PRF efficiency varies with the presence of Annexin A2. Its knockdown significantly increases the frameshifting efficiency whereas its overexpression decreases the PRF efficiency [75]. RG501 is a synthetic compound that has antiviral potential against HIV-1. This compound can bind the upper part of the PRF stem loop [76,77]. The binding stimulates the −1 PRF efficiency and disrupts the stoichiometry between gag/pol proteins. In vitro study using this compound showed a reduced HIV-1 replication in lymphocytic cell lines [78]. However, this compound lacks specificity because it can bind to other RNAs conferring it a toxic profile [77,78].

CRISPR-Cas12a-RNA complex and Antisense oligonucleotides (ASOs) offer innovative approaches to modulate −1 PRF. By mimicking natural pseudoknot structure, CRISPR-Cas12a can form a complex with the target mRNA to disrupt a pseudoknot structure [79]. In vitro and in cell-based studies, this strategy demonstrated more than 50% reduction of −1 PRF in SARS-CoV-2 [79]. Furthermore, the complex specifically binds to the target without inducing RNA cleavage [79]. 

Modified ASOs targeting frameshifting signals have the potential to inhibit PRF with less toxicity than unmodified ASOs [80]. An example is the ASO modified with locked nucleic acids [81]. This oligonucleotide is designed based on the 3D structure of the 3′ pseudoknot. Aligning ASO targeting the stem 1 of the pseudoknot showed a dose-dependent inhibition of the −1 PRF of SARS-CoV-2 in cell-line study. At 100 nM, this ASO successfully inhibited the replication of SARS-CoV-2 [70]. 

Another platinated oligonucleotide modified with a G1·U·G3 triad called 2′-O-methylribooligonucleotide also can inhibit HIV-1 replication. It was designed to align to the 3′ stem loop of HIV-1 PRF forming 2 parallel strands. The alignment induces a rearrangement of the G1, G3 intrastrand crosslink, and then into an interstrand crosslink. Therefore, this process selectively interrupts the translation downstream of the frameshift sequence, reducing the gag/pol ratio [82]. 

### 5.2. Potential of PRF Stimulation

Stimulating PRF efficiency also can be therapeutically beneficial by restoring cellular homeostasis disrupted by disease. This strategy can also alter the stoichiometric equilibrium in retroviruses. 

In cancer, for example, increasing OAZ1 expression through +1 PRF can restrain tumor growth and overcome drug resistance. This potential is exemplified in the use of S11 in cisplatin-resistant cancer cells. S11 is a histone deacetylase inhibitor that enhance histone acetylation in the OAZ1 promoter region [83]. This epigenetic modification promotes OAZ1 expression, which means that it can impact OAZ1 PRF efficiency.

Peptidyl transferase inhibitors can also stimulate PRFs. Sparsomycin, for example, perturbs the peptidyl transfer center by stimulating −1 PRF in L-A virus. The mechanism occurs by stimulating the binding of peptidyl-tRNA to the P-site [74]. 

CRISPR-Cas12a-RNA complex can also be adopted to stimulate −1 PRF. A study in rabbit reticulocyte lysate designed a CRISPR-Cas12a to form a complex with the downstream the slippery sequence. Guided from pseudoknot complex of cardiovirus, the design considers an optimal spacer of 7 nt between the complex and the slippery sequence. Additionally, using a sequence enriched with A-U in the region close to the ribosome enhances the system stability and increases the frameshifting yield. The complex serves as a ribosomal pausing element, triggering a strong ribosome stalling and blocking the elongating ribosome. The complex successfully stimulated viral PRF without inducing RNA cleavage [79]. 

Unlike the above examples, ASOs can induce significant frameshifting events in the absence of natural FSEs. This mechanism could potentially be used to disrupt viral protein stoichiometry. A study demonstrated the potential of assembling RNA oligonucleotides and locked nucleic acid modification. Annealing 12 to 18 nt downstream of the slippery sequence can induce optimal levels of frameshifting by stabilizing the ribosomal tunnel entrance. Locked nucleic acid modification enhances this system by improving stability and binding affinity [80,81,84].

Finally, PRF stimulation also holds valuable potential for mRNA vaccine development, drawing inspiration from the development of predictive indel frameshifts. In this context, the immune system recognizes frameshifting products as neoantigens and shows immune responses [85]. The potential of this technique is exemplified in Lynch syndrome, a hereditary cancer caused by germline mutations. Cancer vaccines formulated with commonly recurring frameshift peptides have shown potential to induce immunogenicity in Lynch syndrome [85]. 

## 6. Discussion and Perspectives

This manuscript reviewed the critical roles of PRF in various biological processes. PRF efficiency impacts gene expression for both normal cellular functions and disease states. PRF efficiency varies significantly across organisms and conditions, influenced by cis-acting elements and trans-acting factors. For instance, PRF can vary from 5% in HIV to 75% in SARS-CoV-2 [35,51]. Understanding the mechanisms governing PRF efficiency is essential for harnessing its potential in therapeutic applications and vaccine development.

One major challenge in targeting PRF is the risk of off-target effects. Approximately 10% of human genes are predicted to possess PRF signals [86,87]. The use of PRF modulators might affect other genes and cause off-target effects. Encouragingly, a comparative study on Paternally expressed gene 10 (PEG10) PRF and HIV PRF showed that their responses toward PRF modulators were different [88]. This suggests the possibility of specifically targeting viral PRF. In addition, the unexpected immune responses from the SARS-CoV-2 IVT mRNA vaccine highlight the challenging impact of PRFs [11]. This concern regarding immunity needs further investigation in order to advance the field of vaccinology. The PRF-induced effect in WNV also shows contradictory results between in vivo and in vitro studies [39,43]. These observations underscore the need for further exploration of PRF.

Another challenge is elucidating the functions of frameshift products. Many of them have ambiguous or unknown roles. In viral PRF, frameshifting proteins counteract host defenses through mechanisms that largely remain unclear. For example, NSP1β in PRRSV influences the production of other viral proteins, suggesting that frameshift proteins play a role in modulating host responses and facilitating infection progression [48]. Similarly, in SARS-CoV-2, the role of NSP12 in circumventing the host immune response has been the focus of several studies [52,55]. However, a consensus on its real function has not been fully established. 

Targeting PRF offers a promising strategy for advancing RNA-based therapies, particularly in vaccine development and in combating viral infections. Addressing challenges including specificity, off-target effect and host-virus interaction is crucial for translating PRF targeting into effective clinical applications. To overcome these barriers and harness the full therapeutic potential of PRF modulation, interdisciplinary efforts and continued innovation are essential.

## Figures and Tables

**Figure 1 ijms-26-01294-f001:**
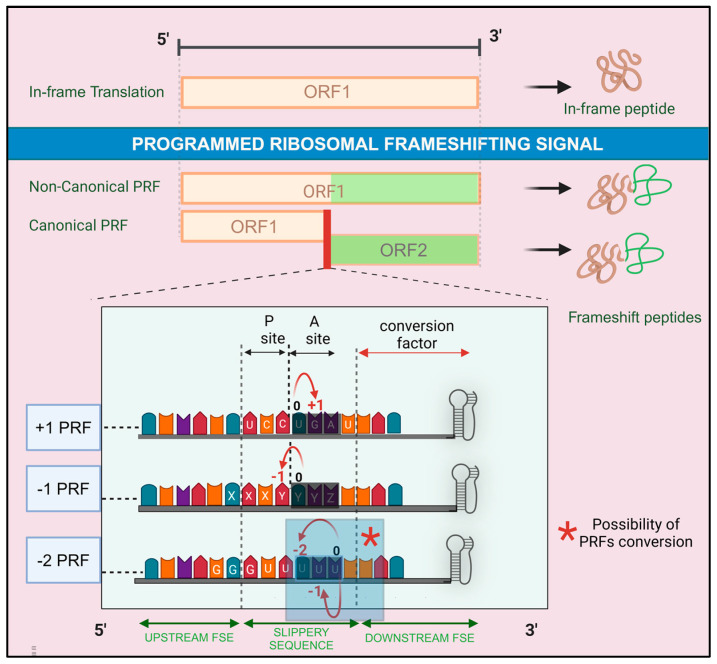
PRF (programmed ribosomal frameshifting) types and mechanisms. PRF occurs when mRNA translation is shifted from the in-frame translation. PRF might be canonical (forming new ORF), or non-canonical (occurring in the same ORF). PRF induces changes in the peptide synthesis. The regulating elements of PRF are usually a slippery sequence, an upstream FSE (frameshift stimulatory element), and a downstream FSE. The downstream FSE often forms pseudoknots. Based on the shifting direction and the number of skipped nucleotides, three types of PRF events exist: +1 PRF which skips one nucleotide from the frame 0 in the 3′ direction. The frame 0 is a stop codon UGA in the example of OAZ1 frameshift. −1 and −2 PRFs shift 1 and 2 nucleotides in the 5′ direction, respectively. For the −1 PRF, the frame 0 is a hungry codon having YYZ pattern. For the −2 PRF, the frame 0 is a UUU stretch. Conversion between PRFs can occur depending on the spacer length between 3′ mRNA pseudoknot and slippery sequence. The −1 and −2 PRFs in arterivirus are shown here. Created in BioRender. Harinirina AIna, R. (2025) https://BioRender.com/z63g739. Accessed on 3 February 2025.

**Figure 2 ijms-26-01294-f002:**
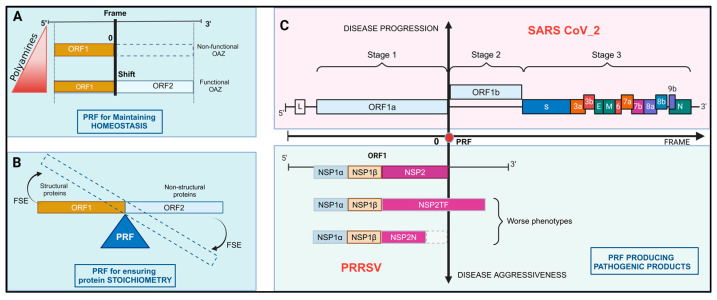
Diverse biological functions of PRF. (**A**) PRF in homeostasis maintenance. In the polyamine metabolism network, a low polyamine level maintains the in-frame translation of OAZ1 mRNA, resulting in a non-functional ORF1 product. High polyamine levels induce +1 PRF to generate a functional OAZ1. (**B**) PRF in maintaining protein stoichiometry. In viruses, PRF ensures the right balance between the amounts of non-structural and structural proteins. (**C**) PRF in producing pathogenic proteins. The diagram represents the mRNA frame in the horizontal line and the disease progression and aggressiveness in the vertical line. The lines junction represents the frame 0. In the example of SARS-CoV-2, ORF1b is the direct product of PRF, leading the infection progression to Stage 2. In the case of Porcine Reproductive and Respiratory Syndrome Virus (PRRSV), non-structural protein 2 (NSP2) variants are induced by PRF. Both the elongated and the truncated NSP2 variants worsen its pathogenicity. Created in BioRender. Harinirina AIna, R. (2025) https://BioRender.com/g52l984. Accessed on 3 February 2025.

**Figure 3 ijms-26-01294-f003:**
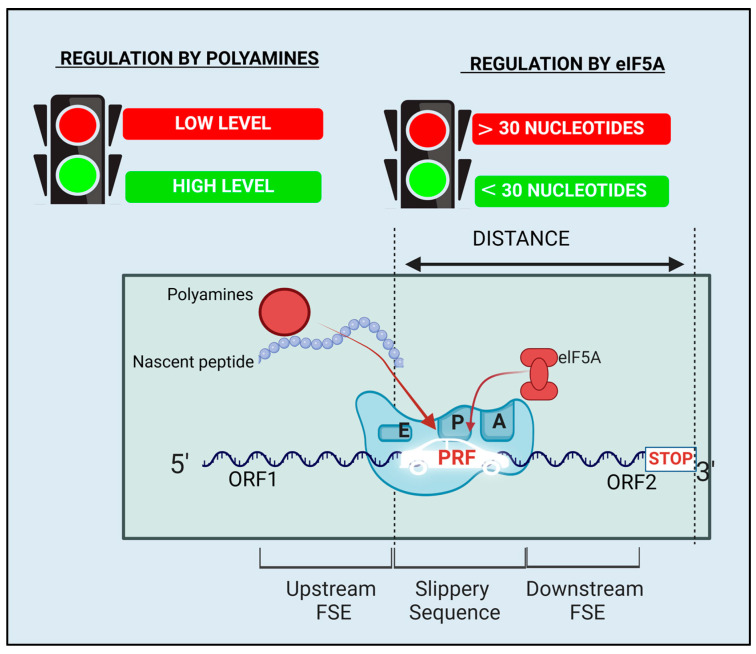
PRF regulation by polyamines and eIF5A. This figure highlights the importance of polyamines levels, eIF5A, and the distance between the in-frame stop codon and the slippery sequence to regulate PRF efficiency [6]. For example, a high level of polyamines ensures the +1 PRF efficiency in the decoding of OAZ1. A nascent peptide within the upstream FSE constitutes the binding site for polyamines regulation. eIF5A is necessary for efficient PRF when the distance between the PRF site and the stop codon are close (approximately 30 nucleotides). eIF5A regulation is impaired for a larger distance. Created in BioRender. Harinirina AIna, R. (2025) https://BioRender.com/r02e305. Accessed on 3 February 2025.

**Figure 4 ijms-26-01294-f004:**
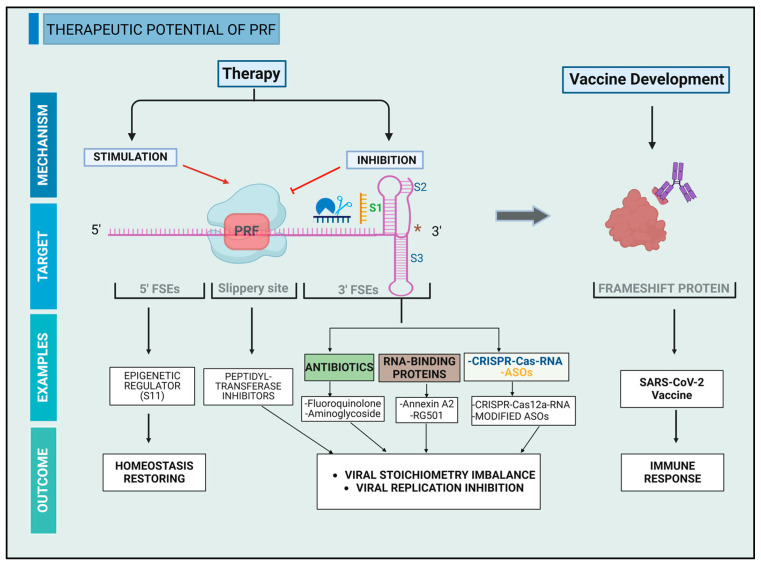
Therapeutic potential of PRF. This figure illustrates the potential of targeting PRF in therapy and vaccine development. In therapy, targeting the 5′ FSE stimulates PRFs, while targeting the 3′FSEs inhibits PRFs. Antibiotics, RNA-binding proteins, CRISPR-Cas-RNA complexes and ASOs bind to specific sequences of the pseudoknot to inhibit PRFs. CRISPR-Cas-RNA complex and ASOs binding can also stimulate PRFs. These binding hold antiviral proprieties by inhibiting viral replication or by disrupting viral protein stoichiometry. The epigenetic regulator S11 has potential to restore homeostasis through stimulating OAZ1 PRF. Peptidyl transferase inhibitors perturb PRF occurrence by acting on the peptidyl transfer center in the core slippery site. Frameshift proteins can induce immune responses, which constitutes the potential of PRF in vaccine development. Created in BioRender. Harinirina AIna, R. (2025) https://BioRender.com/i88x359. Accessed on 3 February 2025. *: Possibility of PRFs conversion.

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
