# Peer review of "Biological Significance and Therapeutic Promise of Programmed Ribosomal Frameshifting"

_ijms, 2025, doi:10.3390/ijms26031294_

Round 1
Reviewer 1 Report
Comments and Suggestions for Authors
This manuscript provides an overview of the interesting topic of programmed ribosomal frameshifting. There are, however, numerous places in the manuscript that lack clarity and depth and could be improved to make the manuscript more impactful to the field. I have summarized my main areas of concern in the following nine points:
Major Points:
1. Introduction 24-36: The first paragraph needs to be polished for accuracy. The points regarding methionine recruitment, triple nucleic acids (three nucleic acids vs three bases in a single mRNA being read, and elongation factors/aminoacyl-tRNAs form a complex with other trans-acting factors can be taken incorrectly if read at face value.
2. Introduction – second paragraph. Rather than any shift in reading frame during translation, a programmed frameshift is a regulated shift in the reading frame that occurs at a designated site within an mRNA. I’m not certain that its appropriate to include translation issues caused by pseudouridine incorporation into mRNA vaccines as a formal programmed frameshift.
3. Fig.1 may be confusing to readers for several reasons. First, shouldn’t the in frame peptide (brown) be part of the resulting peptide from both the canonical (red) and non-canonical (blue PRF. Second, A methionine codon (AUG) is not typically found within a frameshift stimulatory element. Depicting an initiation codon here can also be confusing to general readers. Third, a standard stem loop rather than a more common pseudoknot is pictures as the downstream FSE. Finally, the arrows appear to indicate that frameshifts move more than 1 base for +/-1 or 3 bases for -2.
4. Introduction: line 95 states that -2 PRF is rare and occurs in arterioviruses. I believe that -2PRFs are also seen in HIV and Rous Sarcoma Virus (and likely elsewhere). Thus this statement taken at face value can be misleading.
5. Fig. 2 / line 173Define PRSSV for the reader.
6. Lines 200-201: While altered peptides that are generated can elicit immune responses, I do not believe that there is any evidence to support the statement that the pseudo-U modification impacts vaccine efficacy by altering T cell immunity and B cell antigen stimulation. I believe that this may be an overstatement of the results in that Nature paper (e.g. here’s the last sentence of the abstract from that paper: “although there are no adverse outcomes reported from mistranslation of mRNA-based SARS-CoV-2 vaccines in humans, these data highlight potential off-target effects for future mRNA-based therapeutics and demonstrate the requirement for sequence optimization.”
7. Fig. 3 may also be confusing to readers as it is designed. The upstream FSE appears to be the entire ORF upstream of the frameshift site, and the distance/ration parts of graphical are difficult to directly integrate into the model. I would recommend an extensive redesign of the figure for clarity.
8. Section 5.1/5.2: I would recommend generating a figure to support this section on PRF inhibition/enhancement as well as going into more depth regarding the compounds and proteins that are currently mentioned. This part of the review may be of significant interest and thus should be developed more in my opinion.
9. All sections of the manuscript should be carefully edited to optimize the use of the English language and avoid jargon.
Comments on the Quality of English LanguageAll sections of the manuscript should be carefully edited to optimize the use of the English language and avoid jargon.
Author Response
Major Points:
- Introduction 24-36: The first paragraph needs to be polished for accuracy. The points regarding methionine recruitment, triple nucleic acids (three nucleic acids vs three bases in a single mRNA being read, and elongation factors/aminoacyl-tRNAs form a complex with other trans-acting factors can be taken incorrectly if read at face value.
Thank you for this suggestion. We have revised this part accordingly.
- Introduction – second paragraph. Rather than any shift in reading frame during translation, a programmed frameshift is a regulated shift in the reading frame that occurs at a designated site within an mRNA. I’m not certain that its appropriate to include translation issues caused by pseudouridine incorporation into mRNA vaccines as a formal programmed frameshift.
Thank you for this suggestion. We have revised the information to ensure it is clearly distinguished from other frameshifting events.
- 1 may be confusing to readers for several reasons. First, shouldn’t the in-frame peptide (brown) be part of the resulting peptide from both the canonical (red) and non-canonical (blue PRF. Second, A methionine codon (AUG) is not typically found within a frameshift stimulatory element. Depicting an initiation codon here can also be confusing to general readers. Third, a standard stem loop rather than a more common pseudoknot is pictures as the downstream FSE. Finally, the arrows appear to indicate that frameshifts move more than 1 base for +/-1 or 3 bases for -2.
Thank you for your detailed and constructive suggestions. We appreciate your critical insights and have corrected these caveats in this figure.
- Introduction: line 95 states that -2 PRF is rare and occurs in arterioviruses. I believe that -2PRFs are also seen in HIV and Rous Sarcoma Virus (and likely elsewhere). Thus, this statement taken at face value can be misleading.
Thank you for pointing out this issue. We have rephrased this part.
- 2 / line 173Define PRRSV for the reader.
We have added the full name of PRRSV.
- Lines 200-201: While altered peptides that are generated can elicit immune responses, I do not believe that there is any evidence to support the statement that the pseudo-U modification impacts vaccine efficacy by altering T cell immunity and B cell antigen stimulation. I believe that this may be an overstatement of the results in that Nature paper (e.g. here’s the last sentence of the abstract from that paper: “although there are no adverse outcomes reported from mistranslation of mRNA-based SARS-CoV-2 vaccines in humans, these data highlight potential off-target effects for future mRNA-based therapeutics and demonstrate the requirement for sequence optimization.”
We agree with you on this issue. We revised this part to only focus on the fact that PRF frameshift elicit immune responses.
- 3 may also be confusing to readers as it is designed. The upstream FSE appears to be the entire ORF upstream of the frameshift site, and the distance/ration parts of graphical are difficult to directly integrate into the model. I would recommend an extensive redesign of the figure for clarity.
Thank you for this suggestion. We have changed the whole figure based on your comment.
- Section 5.1/5.2: I would recommend generating a figure to support this section on PRF inhibition/enhancement as well as going into more depth regarding the compounds and proteins that are currently mentioned. This part of the review may be of significant interest and thus should be developed more in my opinion.
Your thorough suggestion is extremely appreciated. We expanded this section and added a new figure (Figure 4).
- All sections of the manuscript should be carefully edited to optimize the use of the English language and avoid jargon.
We carefully double-checked every section of the manuscript to optimize the writing quality and style.

Reviewer 2 Report
Comments and Suggestions for Authors
In this review, the authors described the mechanism of programmed ribosomal frameshifting (PRF) and its biological significance in disease and homeostasis, which will be beneficial to therapeutic strategies based on targeting PRF modulation. The manuscript is well written. Below are some suggestions for revision.
1) Line 78, Reference 15. In this reference, the conversion between +1 PRF and -2 PRF are reported (+1 frameshifting in the mammalian system, in yeast the same frame is reached by -2 frameshifting) and the conversion can be finely regulated (When the length of the spacer between the shift site and the pseudoknot is extended by three nucleotides, +1 and -2 frameshifting become equal). So, it is better to include one paragraph to summarize the conversion between different types of PRF in this section (2. Types and Mechanisms of Programmed Ribosomal Frameshifting)
2) Fig. 1, The numbers are not aligned with the bases.
3) Fig. 3, It is better to replace the “AA” with the three letters abbreviation for amino acid residues of OAZ1 protein.
4) The picture of Graphical abstract: It is also better to replace the “AA” with the single letter abbreviation for amino acid residues of OAZ1 protein. It seems that there should be some other amino acid residues behind AA-AA-AA-Cys-Ser in the “FRAMESHIFT PEPTIDE”. There will be no amino acid residues after AA-Cys-Ser for the non-frameshift peptide.
Author Response
1)Line 78, Reference 15. In this reference, the conversion between +1 PRF and -2 PRF are reported (+1 frameshifting in the mammalian system, in yeast the same frame is reached by -2 frameshifting) and the conversion can be finely regulated (When the length of the spacer between the shift site and the pseudoknot is extended by three nucleotides, +1 and -2 frameshifting become equal). So, it is better to include one paragraph to summarize the conversion between different types of PRF in this section (2. Types and Mechanisms of Programmed Ribosomal Frameshifting)
Thank you for your suggestion. We added another part addressing this conversion.
2) Fig. 1, The numbers are not aligned with the bases.
We have revised this figure.
3) Fig. 3, It is better to replace the “AA” with the three letters abbreviation for amino acid residues of OAZ1 protein.
We have replotted this figure.
4)The picture of Graphical abstract: It is also better to replace the “AA” with the single letter abbreviation for amino acid residues of OAZ1 protein. It seems that there should be some other amino acid residues behind AA-AA-AA-Cys-Ser in the “FRAMESHIFT PEPTIDE”. There will be no amino acid residues after AA-Cys-Ser for the non-frameshift peptide.
Thank you for this suggestion. We have replotted the Graphical Abstract figure.

Round 2
Reviewer 1 Report
Comments and Suggestions for Authors
The authors have adequately addressed the comments raised in the initial round of peer review.
Reviewer 2 Report
Comments and Suggestions for Authors
My concerns have been addressed.